# TNFα Causes a Shift in Gene Expression of *TNFRSF1A* and *TNFRSF1B* Isoforms

**DOI:** 10.3390/genes14051074

**Published:** 2023-05-12

**Authors:** Olga Perik-Zavodskaia, Julia Zhukova, Roman Perik-Zavodskii, Alina Alshevskaya, Julia Lopatnikova, Sergey Sennikov

**Affiliations:** Laboratory of Molecular Immunology, Federal State Budgetary Scientific Institution, Research Institute of Fundamental and Clinical Immunology, 630099 Novosibirsk, Russia; perik.zavodskaia@gmail.com (O.P.-Z.);

**Keywords:** TNFα, *TNFRSF1A*, *TNFRSF1B*, alternative splicing, K562, MCF-7

## Abstract

Alternative splicing is a part of mRNA processing that expands the diversity of proteins encoded by a single gene. Studying the full range of proteins–products of translation of alternatively spliced mRNA is extremely important for understanding the interactions between receptor proteins and ligands since different receptor protein isoforms can provide variation in the activation of signaling pathways. In this study, we investigated the expression of isoforms of TNFR1 and TNFR2 receptors before and after exposure to TNFα in two cell lines that had previously demonstrated diverse effects on cell proliferation under TNFα incubation using RT-qPCR. We found that after incubation with TNFα: (1) expression of isoform 3 of the *TNFRSF1A* gene was increased in both cell lines; (2) the cell line with increased proliferation, K562, had decreased expression of isoforms 1 and 4 of the *TNFRSF1A* gene and expression of isoform 2 of *TNFRSF1B* gene was absent at all; (3) the cell line with decreased proliferation—MCF-7 had significantly increased expression of isoform 2 of *TNFRSF1B* gene. Thus, we can conclude that TNFα exposure to the K562 and MCF-7 cell lines leads to changes in the expression of TNFα receptor isoforms, which, in turn, can appear via diverse proliferative effects.

## 1. Introduction

It is difficult to overestimate the importance of alternative splicing in protein studies. An overwhelming number of multiexon transcripts undergo alternative splicing to provide protein variety. As a result, several protein variations with different characteristics and functions can be produced from a single gene transcript, increasing the complexity of gene expression regulation [1,2]. It was previously shown that there is an effect on cytokine isoform expression under definite stimulation [3]. This means that the expression of different isoforms can be influenced by an external stimulus. It is even possible that the ligand can influence the expression of isoforms of its receptor. However, there is insufficient data on the impact of the ligand on the expression of its receptor isoforms. Research on alternative splicing in the context of the study of receptor proteins will significantly improve our understanding of the complexity of activation and regulation of signaling pathways, provide clues to understanding the etiology of diseases, and help in developing therapies.

By interacting with its receptors TNFR1 and TNFR2, the tumor necrosis factor α (TNFα) ligand is involved in the inflammatory response activation, cell proliferation, and differentiation [4]. TNF-α receptor expression levels have been shown to change significantly in immunopathological conditions of various etiologies, such as bronchial asthma [5] and rheumatoid arthritis [6], with the severity and duration of the disease also affecting the distribution of receptor expression on immunocompetent cells [7].

Different studies demonstrated that TNFR1 and TNFR2 receptor isoforms have variations in response to TNF-α stimulation and anti-TNF-α therapy [4,8,9]. However, despite the accumulated data on the peculiarities of functioning and regulation of TNFR1 and TNFR2 signaling pathways, they remain incompletely understood.

Previous studies of TNFα-stimulated cell lines performed in our laboratory demonstrated a polar dose-dependent change in the proliferative potential of MCF-7 and K-562 cell lines both towards its increase (MCF-7 line) and towards its decrease (K-562 line) [10]. We suggested that this difference may be associated with the expression of different splice variants of TNFα receptor genes (*TNFRSF1A* and *TNFRSF1B*). Only proteins with a signal peptide in their amino acid sequence can become receptor proteins [11]. Thus, this study aims to determine the expression of signal peptide-positive splice variants of TNFR1 and TNFR2 receptors in MCF-7 and K-562 cell lines upon stimulation with different doses of TNFα.

## 2. Materials and Methods

### 2.1. Signal Peptide-Positive Protein Coding Isoform Selection via InterPro

In this study, we examined mRNA isoforms of *TNFRSF1A* and *TNFRSF1B* genes, the sequences of which we took from the NCBI database: RefSeq RNA *TNFRSF1A*—https://www.ncbi.nlm.nih.gov/nuccore?LinkName=gene_nuccore_refseqrna&from_uid=7132 (accessed on 7 March 2023) and RefSeq RNA *TNFRSF1B*—https://www.ncbi.nlm.nih.gov/nuccore?LinkName=gene_nuccore_refseqrna&from_uid=7133 (accessed on 7 March 2023). We found open reading frames (ORF) in the nucleotide sequences of isoforms in Unipro UGENE and analyzed the ORF data using the InterPro neural network for the presence of the signal peptide using the InterPro web server https://www.ebi.ac.uk/interpro/search/sequence/ (accessed on 7 March 2023). Only isoforms with the signal peptide were selected for further analysis.

### 2.2. Primer Design

We performed multiple sequence alignment of all isoforms’ nucleotide sequences for both *TNFRSF1A* and *TNFRSF1B* genes using MUltiple Sequence Comparison by Log-Expectation (MUSCLE) in UniPro UGENE. As *TNFRSF1A* and *TNFRSF1B* genes are not well annotated, we selected primers unique to the isoforms (and not to the specific intron or exon) and validated them using the web servers Oligocalculator—http://www.bio.bsu.by/molbiol/oligocalc.html (accessed on 7 March 2023)—and Multiple Primer Analyzer—https://www.thermofisher.com/ru/ru/home/brands/thermo-scientific/molecular-biology/molecular-biology-learning-center/molecular-biology-resource-library/thermo-scientific-web-tools/multiple-primer-analyzer.html (accessed on 7 March 2023). TNFRSF1A isoform 4 is labeled as a nonsense-mediated decay transcript at NCBI; however, it contains a weak Kozak sequence (TCTCT), an open reading frame, and a signal peptide sequence, so there is a possibility for it to be a regular protein-receptor-coding transcript as well. It was not possible to design primers for *TNFRSF1B* isoform 1; therefore, we selected primers for the constant region of *TNFRSF1B* to indirectly assess the effect of other isoforms (1, 4, and 5). The designed primers are presented in Table 1, Table 2 and Table 3. Primed binding sites are shown in Figure 1.

### 2.3. Cell Line Culturing with TNFα

We used MCF-7, an epithelial-like cell line derived from invasive adenocarcinoma of human mammary glands, and K-562, an erythromyeloblastoid cell line derived from human myeloid leukemia cells. Previous studies have demonstrated that for the MCF-7 cell line, TNFα stimulation at 5 ng/mL, and for the K562 cell line, TNFα stimulation at 25 ng/mL, resulted in differently directed changes in the proliferation of these cell lines (K562 increased proliferation, MCF-7 decreased proliferation); so, these are the concentrations used in this present study [10]. The incubation of cell lines with TNFα lasted 72 h, and then the cells were harvested for total RNA isolation.

### 2.4. Total RNA Extraction

We isolated total RNA from cells using the Total RNA Purification Plus Kit (Norgen Biotek, Thorold, ON, Canada, 48,400); we also isolated the measured concentration of the RNA on the NanoDrop 2000c. We froze the total RNA at −80 °C until the reverse transcription of the RNA was achieved.

### 2.5. Reverse Transcription

We performed reverse transcription of the total RNA samples (*n* = 4) using RNAscribe RT and oligo-dT primers (Biolabmix, R04-50, Novosibirsk, Russia). We used an input of 100 ng of the RNA. Reverse transcription was conducted as follows: 55 °C for 50 min and 85 °C for 10 min.

### 2.6. PCR and Melt Curve Analysis

We performed touchdown qPCR with melt curve analysis using UDG HS-qPCR Lo-ROX SYBR (×2) Mix (Biolabmix, MHR033-2040), 1 μL out of the 20 μL of RT product, Table 1, Table 2 and Table 3 primers at a final concentration of 500 μM. All reactions were conducted using four technical replicates. Touchdown qPCR and melt curve analysis were conducted as follows: 55 °C for 2 min; 95 °C for 5 min; 6 cycles of 95 °C for 20 s, 66 °C -> 60 °C for 30 s (1 °C/cycle decrement), and 72 °C for 1 min, respectively; 34 cycles of 95 °C for 15 s, 60 °C for 20 s, and 68 °C for 1 min, respectively; 72 °C for 5 min; melt curve of 95 °C -> 65 °C (1 °C/step decrement).

Melt curves’ temperatures matched the predicted amplicon melting temperatures, thus ensuring amplicon specificity (Figure 2).

### 2.7. Relative Isoform Expression Analysis

The relative expression level of *TNFRSF1A* and *TNFRSF1B* gene isoforms was calculated using the method of Taylor et al. [12]. In brief, Ct values for each isoform and housekeeping gene were obtained via quantitative PCR. Then, we calculated the relative expression of each isoform; Ct (Threshold Cycle) values of the target gene isoforms were normalized using the mean of the geometric means of the used housekeeping genes.

### 2.8. Differential Isoform Expression Testing

We used multiple *T*-tests with FDR correction for differential isoform expression testing in GraphPad Prism 9.4 for macOS. We considered *q*-values < 0.05 significant.

## 3. Results

### 3.1. Analysis of Isoforms of TNFRSF1A and TNFRSF1B Genes for the Presence of Signal Peptide in Their Amino Acid Sequence

We used the InterPro neural network on all *TNFRSF1A* and *TNFRSF1B* isoforms and found a signal peptide in isoforms 1, 3, and 4 of the *TNFRSF1A* gene and in isoforms 1, 2, and 3 of the *TNFRSF1B* gene. We chose all of the above-mentioned isoforms for further experiments as all of them contained an open reading frame and a signal peptide, except isoform 1 of the *TNFRSF1B* gene (due to the technical impossibility of primer design). Regarding exon/intron composition, we noted the following: *TNFRSF1A* isoform 1 contains exons 1–5, exons 7–10, intron 9, and exon 11; *TNFRSF1A* isoform 3 contains exons 1–10, intron 9, and exon 11; *TNFRSF1A* isoform 4 contains exon 1, exons 3–5, exons 7–10, intron 9, and exon 11; *TNFRSF1B* isoform 2 contains exon 1 and exons 4–11; *TNFRSF1B* isoform 3 contains exons 3–11.

### 3.2. Study of TNFRSF1A and TNFRSF1B Gene Isoform Expression after TNFα Exposure

We incubated cell lines K562 and MCF-7 with TNFα, evaluating changes in the expression of receptor isoforms for stimulated and intact cultures. For the K562 cell line stimulated with TNFα, we found a decrease in the expression of isoforms 1 and 4 of the *TNFRSF1A* gene and an increase in the expression of isoform 3 in response to the stimulation (Figure 1).

For the K562 cell line, no statistically significant differences were found in the expression of TNFRSF1B isoform 3, and TNFRSF1B isoform 2 was not detected at all (Figure 2).

For MCF-7 cells, no significant changes in the expression of TNFRSF1A isoforms 1 and 4 were found; however, a statistically significant increase in the expression of isoform 3 was shown in response to the TNFα stimulation of cells (Figure 3).

We observed an increase in the expression of isoform 2 of the *TNFRSF1B* gene in the MCF-7 cell line in response to TNFα stimulation. No statistically significant change in expression was observed for isoform 3 of this gene (Figure 4).

## 4. Discussion

As a result of this study, we evaluated the expression of the splice variants of *TNFRSF1A* and *TNFRSF1B* receptor-encoding genes for TNFα on tumor cell lines of different origins in intact and TNFα-stimulated cultures.

An increased expression of isoform 3 of the *TNFRSF1A* gene was detected for both cell lines, epithelial-like (K562) and myeloblast-like (MCF-7). The difference between the cell lines comes from the decreased expression of isoforms 1 and 4 of the *TNFRSF1A* gene in K562, while no changes were found in MCF-7.

For the *TNFRSF1B* gene, we found significantly increased expression of isoform 2 for the MCF-7 cell line, whereas no expression of this isoform was found at all in the K562 cell line.

We believe that these differences in the isoforms’ expression profiles of K562 and MCF-7 cell lines may be responsible for the differentially directed functional responses described previously [10], which highlights the role of alternative splicing of receptors in the regulation of cell biological response under the impact of stimulating ligand concentrations. It must be noted that the treatment of MCF-7 cells with TNFα exposes latent ERα receptors, as reported by Franco et al. [13]. This suggests that the signaling of estrogens through ERα receptors may potentially influence the differential expression of our studied isoforms. However, more research is still required to deepen our understanding of the relationship between alternative splicing of receptor transcripts, the functional response of cells, and the ability to manipulate these processes for potential therapies.

## Data Availability

The data that support the findings of this study are available from the corresponding author, S.V. Sennikov, upon an email request.

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
