# Peer review of "TNFα Causes a Shift in Gene Expression of *TNFRSF1A* and *TNFRSF1B* Isoforms"

_genes, 2023, doi:10.3390/genes14051074_

Round 1
Reviewer 1 Report
The work by Olga Perik-Zavodskaia et al. aimed to study the changes in the expression profile of TNF-alpha receptors in two cancer cell lines under the action of TNF-alpha. The authors demonstrated a shift in mRNA quantities after stimuli. The logic flow of the manuscript is clear. However, major revisions are necessary
Major issues.
1. The work was done using only two cell lines and no animal models was used to prove the results. How specific are these results to tumor and normal cells?
2. The conclusions are made based on only real-time RT‒PCR results which detects only mRNA. Western blotting or any other assay must be used to verify the changes on protein level.
3. The analysis of isoform expression must be accompanied by the analysis of total mRNA of the gene. The changes in isoform quantities may be the result of the changes in total mRNA and not in the proportion of isoforms.
4. The work lacks functional tests for cells with different expression of isoforms., i.e., the authors demonstrated the changes in isoform expression, but what is the consequence for the cells?
Minor issues
1. The term isoform is used for protein. For mRNA it appropriate to use splice-variant, which is actually may not by translated.
2. There is a typo in line 42.
3. The method of relative expression of the isoforms (line 111) must be described briefly.
The manuscript is not publishable at the present state. Major revision is necessary.
Author Response
We express our sincere gratitude for your valuable feedback on our research work! We carefully considered your constructive comments, and we have provided detailed explanations for our experimental design choices.
We hope that our responses to your questions and comments have been satisfactory, and we hope you consider our article for publication!
Major issues response:
1st - The work was done using only two cell lines and no animal models were used to prove the results. How specific are these results to the tumor and normal cells?
To answer your question, we need to clarify the primary hypothesis of the study. Previous research conducted in our laboratory on TNFα-stimulated cell lines showed a polar dose-dependent change in the proliferative potential of MCF-7 and K-562 cell lines, with an increase in the case of MCF-7 and a decrease in the case of K-562. Based on this data, we hypothesized that the difference in response may be due to the activation of different isoforms of receptor genes to their ligand. Therefore, our choice of these cell lines was based on the presence of the required characteristics that allowed us to test this hypothesis.
Specifically, we needed cell lines that expressed both variants of the receptor, which at the same time demonstrated an opposite dose-dependent effect on the proliferation of cells. Thus, our goal was not to study the effect of TNFα specifically on these tumor cell lines but to study the molecular mechanism of ligand-induced changes in the expression of alternatively spliced genes of its receptors.
However, we fully agree that it is necessary to expand the range of cell lines used and test the data on animal models.
This study was thought to be a probing study. As we have now observed the possibility of differential isoform expression in response to TNF, we plan on testing the same in human cells using spatial transcriptomics.
2nd - The conclusions are made based on only real-time RT‒PCR results which detect only mRNA. Western blotting or any other assay must be used to verify the changes in protein level.
We agree that Western blotting and other protein-level assays can be valuable tools for validating changes observed at the mRNA level. However, we believe that the conclusions drawn from our study are well-supported by the real-time RT-PCR results that we obtained. It is important to note that the goal of our study was to investigate changes in gene expression levels, not changes in protein expression levels.
The Proteomics study of isoforms can be an interesting continuation of this study!
3rd - The analysis of isoform expression must be accompanied by the analysis of the total mRNA of the gene. The changes in isoform quantities may be the result of the changes in total mRNA and not in the proportion of isoforms.
We did just that with TNFRSF1B! In Figure 4 we can see, the total expression gene and isoform-specific expression: TNFRSF1B and its Isoform 2 saw an increase in gene expression while Isoform 3 had no statistically significant changes. So we can say that isoform-specific gene expression is not dependent on the changes in total mRNA. Differences in total mRNA are also eliminated by the housekeeping genes normalization – cells with drastically different total RNA quantities would still be adequately measured in terms of gene expression.
4th - The work lacks functional tests for cells with different expression of isoforms., i.e., the authors demonstrated the changes in isoform expression, but what is the consequence for the cells?
We simply did functional analysis before we did gene expression studies! The selection of MCF-7 and K-562 cell lines was based on data from previous studies in which we carefully tested the functional response of different cell cultures in response to different concentrations of TNF-alpha, and because we found that these cell lines exhibit a polar dose-dependent change in proliferative potential, they were selected for further work.
Minor issues response
1st - The term isoform is used for protein. For mRNA it is appropriate to use splice-variant, which may actually not be translated.
We acknowledge that the term "isoform" is commonly used to refer to protein variants resulting from alternative splicing of mRNA. However, the term "isoform" has also been used in the literature to refer to mRNA variants resulting from alternative splicing, even the NCBI database uses it as a default term along with the “transcription variant” term.
2nd - There is a typo in line 42.
We used British English in our manuscript, and the terminology we employed is appropriate for British English. However, to alleviate any potential confusion, we have revised the manuscript to conform to American English conventions.
3rd - The method of relative expression of the isoforms (line 111) must be described briefly.
We added the description: In brief, Ct values for each isoform and housekeeping gene were obtained by quantitative PCR. Then, we calculated the relative expression of each isoform: Ct (Threshold Cycle) values of the target gene isoforms were normalized using the mean of the geometric means of the used housekeeping genes.
Reviewer 2 Report
1. Why this choice of cell lines at all? They are widely different in background and general phenotype, one being derived from mammary epithelium, while the other is a lymphoblast.
2. Why use different concentrations of TNF for the stimulation. The design is based off a previous paper from the same group where a range of TNF-α concentrations is employed and changes in the receptor expression are verified. But the previous paper also doesn’t use the same concentrations for different cell lines. At the very least both 25 and 5 ng/mL concentrations should be used for both cell lines.
3. An additional problem is that MCF-7 is an estrogen responsive cell line and TNF-α is capable of cross-activation of the ER-α. Ideally, a cell line lacking estrogen receptors, such as MDA-MB-231 would be used to eliminate any effects of TNF through ER activation.
The authors should consider adding an additional cell line to experiments and perform them using the same TNF concentrations. As it is, there is no way to compare the results obtained for the different cell lines.
Author Response
We express our sincere gratitude for your valuable feedback on our research work! We carefully considered your constructive comments, and we have provided detailed explanations for our experimental design choices.
We hope that our responses to your questions and comments have been satisfactory, and we hope you consider our article for publication!
Major issues response:
1st - Why this choice of cell lines at all? They are widely different in background and general phenotype, one being derived from mammary epithelium, while the other is a lymphoblast.
To answer your question, we need to clarify the primary hypothesis of the study. In past studies in our laboratory, we have tested in various cell lines the effects of TNF-alpha on altering the density of the two TNFR1 and TNFR2 receptors (non-isoforms), their co-expression (at the protein level) and their proliferative potential. Research showed a polar dose-dependent change in the proliferative potential of MCF-7 and K-562 cell lines, with an increase in the case of MCF-7 and a decrease in the case of K-562. Based on this data, we hypothesized that the difference in response may be due to the activation of different isoforms of receptor genes to their ligand.
Therefore, our choice of these cell lines was based on the presence of the required characteristics that allowed us to test this hypothesis. Specifically, we needed cell lines that expressed both TNFR1 and TNFR2 receptors, which at the same time demonstrated an opposite dose-dependent effect on the proliferation of cells. Our goal was to study the molecular mechanism of ligand-induced changes in the expression of alternatively spliced genes of its receptors.
2nd - Why use different concentrations of TNF for the stimulation. The design is based off a previous paper from the same group where a range of TNF-α concentrations is employed and changes in the receptor expression are verified. But the previous paper also doesn’t use the same concentrations for different cell lines. At the very least both 25 and 5 ng/mL concentrations should be used for both cell lines.
Indeed, in our previous study, the same concentrations were used for all cell lines because the aim of the study was to see which concentrations could lead to a noticeable cellular response and a change in receptor expression, which logically implies a change in the transcriptional profile.
We agree that using the full range of TNF-alpha concentrations for both cell lines would have been ideal and interesting in view of the diverse and extensive accumulation of data.
But because of experimental limitations, we had to choose the concentrations at which we observed the greatest cell response. And the greatest response for K562 and MCF-7 cell lines were found at concentrations of 5 and 25 ng/mL, respectively.
3rd - An additional problem is that MCF-7 is an estrogen responsive cell line and TNF-α is capable of cross-activation of the ER-α. Ideally, a cell line lacking estrogen receptors, such as MDA-MB-231 would be used to eliminate any effects of TNF through ER activation.
Thank you for underlying this limitation of our study, we really appreciate it! Indeed MCF-7 cells are estrogen-responsive and TNF-alpha can potentially cross-activate ER-α.
The MDA-MB-231 cell line could be a better option for our future experiments aimed at studying the effect of TNF-α on TNFR1 and TNFR2 activation. We appreciate the suggestion!
We have pointed out the information you gave as a research limitation in the manuscript text: “It must be noted that the treatment of MCF-7 cells with TNFα exposes latent ERα receptors, as reported in Franco et al. [13]. This suggests that the signaling of estrogens through ERα receptors may potentially influence the differential expression of our studied isoforms.“
Reviewer 3 Report
In the present study the authors investigated the expression of isoforms of TNFR1 and TNFR2 receptors before and after exposure to TNFα in two cell lines (K562 and MCF-7). Based on their findings, they concluded that TNFα exposure can cause changes in the expression of TNFα receptor isoforms in these cell lines, and that these changes may have different effects on cell proliferation.
A search on Ensembl or UCSC Genome Browsers for the splice variants of the genes encoding TNFR1 and TNFR2 receptors (TNFRSF1A and TNFRSF1B, respectively) shows that they have several potential protein coding variants, but there is not experimental evidence available for all of them. To identify the isoforms with potential to encode membrane receptors the authors searched for open reading frames (ORF) in the nucleotide sequences of the isoforms and looked for the presence of the signal peptide with the argument that only proteins with a signal peptide in their amino acid sequence can become receptor proteins. Based on this analysis they designed primers to detect mRNA isoforms by RT-qPCR.
Major issues:
1) The links provided in section 2.1 of the Materials and Methods to access the sequences of TNFRSF1A and TNFRSF1B in NCBI are not valid.
2) The description of the different isoforms should be provided. Which exons are included in each of the isoforms?
3) A scheme of the different isoforms highlighting the region of hybridization of the primers used to specifically detect them should be included.
4) Most importantly appropriate controls must be included to guarantee specificity of the primer for the isoforms.
5) Isoform 4 of TNFRSF1A (NR_144351.2) is described in NCBI as a non-coding variant as the transcript is a candidate for nonsense-mediated mRNA decay (NMD). However, the authors included this isoform in their analysis because they found it contains a signal peptide and therefore has potential to encode a membrane receptor. What is the reason for this inconsistency?
Overall, this is an interesting study but without proper controls its relevance is limited.
Author Response
We appreciate your careful review of our work and are glad to hear that you find our research to be valuable!
Major issues response:
1st - The links provided in section 2.1 of the Materials and Methods to access the sequences of TNFRSF1A and TNFRSF1B in NCBI are not valid.
We checked this using multiple PCs and Macs - the links are still accessible! We added a database name for manual lookup as well – for example, RefSeq RNA TNFRSF1A.
2nd - The description of the different isoforms should be provided. Which exons are included in each of the isoforms?
Thank you for your comments and questions regarding our study on TNF-alpha receptor isoforms. We agree that it is important to provide a clear description of the different isoforms and the exons included in each. Unfortunately, TNFR1 and TNFR2 isoforms are not characterized well and so we cannot provide You and Others with the exact details.
We used isoform-specific sequences to pick the primers, which provided isoform specificity.
3rd - A scheme of the different isoforms highlighting the region of hybridization of the primers used to specifically detect them should be included.
We added this information in Scheme 1.
Scheme 1. Primers’ binding sites and amplicons.
4th - Most importantly appropriate controls must be included to guarantee specificity of the primer for the isoforms.
Thank you for your valuable feedback on our research. We agree that including appropriate controls to ensure the specificity of the primers is crucial to the validity of our results. In our study, we performed melt curve analysis to verify the specificity of the amplification products.
We added melt curve analysis graphs (Scheme 2). The good thing is that they correspond to the predicted melting temperature thus ensuring target specificity. We also added amplicon melting temperatures to the primer in Tables 1 and 2.
Scheme 2. Melt curve graphs of the studied TNFRSF1A and TNFRSF1B genes’ isoforms.
5th - Isoform 4 of TNFRSF1A (NR_144351.2) is described in NCBI as a non-coding variant as the transcript is a candidate for nonsense-mediated mRNA decay (NMD). However, the authors included this isoform in their analysis because they found it contains a signal peptide and therefore has potential to encode a membrane receptor. What is the reason for this inconsistency?
We appreciate Your attention to detail and would be glad to explain our criteria. We understand that NCBI describes this isoform as a non-coding variant and a candidate for nonsense-mediated mRNA decay (NMD), however, TNFRSF1A isoform 4 contains a Kozak sequence (TCTCT), an open reading frame, and a signal peptide sequence, so it can be a regular protein-receptor-coding transcript as well! The field of RNA biology can be controversial, but we believe we used solid criteria for this pick.

Round 2
Reviewer 1 Report
The authors addressed all my initial concerns. I recommend to accept the manuscript.
Author Response
We would like to thank you for Your Review, it has helped us a Great Deal to improve our manuscript!
Reviewer 2 Report
I would like to thank the authors for taking the time to review their manuscript and provide changes and answers. The provided answers were satisfactory.
Author Response
We would like to thank you for Your Review, it has helped us a lot to improve our manuscript!
Reviewer 3 Report
The authors have revised the manuscript in accordance with the comments, addressing the major concerns raised. The links provided in section 2.1 of the Materials and Methods to access the sequences of TNFRSF1A and TNFRSF1B in NCBI are now working. A scheme highlighting the region of hybridization of the primers used to specifically detect the different RNA isoforms was included. However, a description of the different RNA isoforms indicating which exons are included in each of the isoforms was not provided. The authors claim that TNFR1 and TNFR2 isoforms are not well characterized. To address the specificity of the primer for the different isoforms the authors included melt curve analysis to verify the specificity of the amplification products.
Author Response
We would like to thank you for Your Review, it has drastically helped us to improve our manuscript!
We were finally able to find the correct nomenclature for the studied isoforms and added their description to the Results Section 3.1: «TNFRSF1A isoform 1 contains exons 1-5, exons 7-10, intron 9, exon 11; TNFRSF1A isoform 3 contains exons 1-10, intron 9, exon 11; TNFRSF1A isoform 4 contains exon 1, exons 3-5, exons 7-10, intron 9, exon 11; TNFRSF1B isoform 2 contains exon 1, exons 4-11; TNFRSF1B isoform 3 contains exons 3-11.»